# Strategic Challenges to the Eradication of African Swine Fever Genotype II in Domestic Pigs in North Italy

**DOI:** 10.3390/ani14091295

**Published:** 2024-04-25

**Authors:** Silvia Pavone, Silvia Bellini, Carmen Iscaro, Marco Farioli, Mario Chiari, Antonio Lavazza, Luigi Ruocco, Davide Lelli, Giorgia Pintus, Paola Prati, Francesco Feliziani

**Affiliations:** 1National Reference Laboratory for Pestivirus and Asfivirus, Istituto Zooprofilattico Sperimentale dell’Umbria e delle Marche “Togo Rosati” (IZSUM), Via G. Salvemini, 1, 06126 Perugia, Italy; c.iscaro@izsum.it (C.I.); f.feliziani@izsum.it (F.F.); 2Istituto Zooprofilattico della Lombardia ed Emilia-Romagna, Via A. Bianchi 7/9, 25124 Brescia, Italy; silvia.bellini@izsler.it (S.B.); antonio.lavazza@izsler.it (A.L.); davide.lelli@izsler.it (D.L.); paola.prati@izsler.it (P.P.); 3Regione Lombardia UO Veterinaria Direzione Generale Welfare, Piazza Città di Lombardia, 1, 20124 Milano, Italy; marco_farioli@regione.lombardia.it (M.F.); mario_chiari@regione.lombardia.it (M.C.); 4Ministero della Salute Direzione Generale della Sanità Animale e del Farmaco Veterinario, Ufficio III Sanità Animale e Gestione Operativa del Centro Nazionale di Lotta ed Emergenza Contro le Malattie Animali e Unità Centrale di Crisi, Viale Giorgio Ribotta, 5, 00144 Roma, Italy; l.ruocco@sanita.it; 5Local Health Authority (ATS), Via Indipendenza, 3, 27100 Pavia, Italy; giorgia_pintus@ats-pavia.it

**Keywords:** African Swine Fever, control, domestic pigs, epidemic disease, epidemiology, eradication, Italy, management, surveillance, wild boars

## Abstract

**Simple Summary:**

African swine fever (ASF) is a highly lethal viral disease affecting suids and caused by the African Swine Fever Virus (ASFV). ASF was described for the first time in 1921 in Kenya. The ASF genotype I virus was introduced to Europe in 1957, marking the onset of the first European wave. In 2007, ASFV genotype II was detected in Georgia, affecting domestic pigs and wild boars, and it later spread to several European and extra-European countries, including Italy. This report focuses on the strategic challenges encountered in the attempt to eradicate ASFV amongst domestic pigs in the Lombardy region. The joint efforts that were implemented facilitated the eradication of ASF in domestic pigs in just over 1.5 months, representing an example of effective and timely cooperation to mitigate both the spread of the infection and the economic repercussions for the Italian and global pig industries.

**Abstract:**

African swine fever (ASF) is a severe viral disease characterized by high lethality in suids and caused by the African Swine Fever Virus (ASFV). The ASF genotype I virus was introduced to Europe in 1957, marking the onset of the first European epidemic wave. In 2007, ASFV genotype II was detected in Georgia, affecting domestic pigs and wild boars before spreading to various European and extra-European countries, including Italy. The first case of ASFV in Italy was documented on 7 January 2022, in a wild boar in the Piedmont region. Since then, several ASFV-positive wild boar carcasses have been identified in the Piedmont and Liguria regions. By June 2023, ASFV had spread to Lombardy, one of the major pig-producing regions in northern Italy; the virus was first detected in early summer in wild boar carcasses. Two months later, it was diagnosed in a commercial pig farm as a consequence of the disease’s spread amongst wild boars and an increase in the viral environmental load. This report aims to describe the features of ASFV domestic pig outbreaks that occurred in the Zinasco municipality (Lombardy) and the joint efforts to mitigate potential direct and indirect economic impacts on the Italian and global pig industry. The epidemiological investigation and the measures implemented, which were all performed according to national and European regulations, as well as exceptional ad hoc measures aimed at protecting the pig industry, are described in order to provide a practical and effective approach to combating ASF.

## 1. Introduction

African swine fever virus (ASFV) is a large DNA virus of the Asfarviridae family and genus Asfavirus [1,2]. ASFV causes a severe disease with exceptionally high lethality in suids (domestic pigs and Eurasian wild boar (Sus scrofa)) characterised by haemorrhagic fever when virulent strains are involved [2]. Notably stable, it may resist a wide range of pH values, temperatures (even remaining infectious for years in frozen meat), and autolysis, remaining infectious for several weeks in carcasses [3]. Cooking (60 °C for 30 min) [4], extremely acidic or basic pH values (pH < 3.9 or >11.5 in serum-free medium), and specific chemical compounds [5] can effectively inactivate it. To date, 24 virus genotypes (I–XXIV) have been defined based on the amplification and sequencing of the variable 3′-end of the B646L gene that encodes p72, which is the main capsid protein [6].

ASF was described for the first time in 1921 in Kenya [7]. It currently involves more than 30 African countries, threatening the swine industry in Sub-Saharan Africa [6]. The ASF genotype I virus was introduced to Europe in 1957, generating the first European wave [8]. The disease was successfully eradicated in all countries except Sardinia, where it has been present since 1978 [9]. In 2007, ASFV genotype II reached Georgia, affecting domestic pigs and wild boars [10], and quickly spread to other countries, becoming a global threat [11,12,13,14,15,16,17,18,19,20,21,22,23,24]. The epidemiological cycle involving the Eurasian wild boar (*Sus scrofa*), known as the wild boar–habitat cycle [5], represents a significant challenge in combating ASF in these territories. A growing population of wild boars, which serve as the main reservoir of ASFV in the environment, increases the chances of the virus spreading into new geographical areas and persisting long-term in the wild, thereby heightening the risk of its introduction into the domesticated swine population [25]. A combination of control measures, including biosecurity measures, fencing, sniper-guided shooting and trapping of wild boars, along with the crucial task of searching for and disposing of wild boar carcasses, tailored to the epidemiological situation, is currently considered effective for ASF control [26]. However, variations among countries in landscape, farming practices (free-range or semi-free-range versus intensive farming), and the level of biosecurity measures applied to swine farms may influence the management of the disease and the success of implemented actions.

In Italy, ASFV genotype II was first detected in January 2022 [27], initially involving only wild boars in northwest Italy. In particular, on 7 January 2022, the first case was reported in the municipality of Ovada (Alessandria Province), a small town in the south of the Piedmont region that is close to the border of the Liguria region. On 7 January 2022, the second case of ASF was reported in a wild boar found agonising in a forest in Fraconalto, another small town of Alessandria Province. On 8 January 2022, in Isola del Cantone (Genoa Province), Liguria region (about 30 and 40 km, respectively, away from the first two cases in Piedmont Region, respectively) a third case was detected, making this a bi-regional cluster of infection [9]. In Italy, as in other European countries, wild boar populations have experienced significant expansion resulting in the colonization of new habitats, such as agricultural lands, peri-urban, and urban areas [28]. The Italian Apennines showed a high wild boar population, ranging from four to eighteen animals per km^2^ [28]. Specifically, the local wild boar density in Liguria and Piedmont was estimated to be 60,000 and 85,000, respectively [9]. Prompt implementation of any measures required by the national and international regulations necessary to avoid the spread of the disease was applied, as described in a previously published study [9]. However, the unique landscape characterised by rocky terrain, rivers, and bridges, encompassing urban areas and bordered by rural and forested areas, coupled with the lengthy bureaucratic processes in Italy, impeded the construction of effective fences to halt the spread of ASF. On the other hand, a specific decree detailing generic biosecurity measures for swine farms was implemented on 28 June 2022 to protect the pig sector [29]. Furthermore, reinforced biosecurity measures were applied in infected zones and in restricted zone I, II, and III according to the provisions outlined in annex III of the Commission Implementing Regulation (EU) 2023/594 [30].

Notwithstanding these control actions, the virus spread to neighbouring areas; on 21 June 2023, the veterinary services collected tissue samples from a wild boar found dead in the municipality of Torretta di Bagnaria (Pavia Province), Lombardy Region. The real-time PCR for ASFV performed by the regional animal health laboratory—the Istituto Zooprofilattico Sperimentale della Lombardy e dell’Emilia Romagna (IZSLER)—showed a positive result. This result was confirmed by the National Reference Laboratory (NRL) for ASFV on 23 June 2023.

This report aims to describe a complex ASFV domestic pig outbreak cluster in the Zinasco municipality (Pavia province, Lombardy region, Italy) stemming from the disease’s spread amongst wild boars and the increase in the viral environmental load, while also emphasizing collaborative efforts to mitigate potential direct and indirect economic impacts on the Italian and global pig industry. A detailed epidemiological investigation and joint actions put in place by Veterinary Services, Regional Veterinary Public Health and Food Safety, Central Crisis Unit (CCU), Group of Experts (GOE), Carabinieri NAS (Unit for Health Protection), and other institutions are described. Sharing the Italian management experiences in tackling the critical and emergency situation might provide a practical approach to combating the devastating consequences of an outbreak in domestic pigs not promptly identified.

## 2. Case Description

### 2.1. Context Data

The ASFV infection cluster that started on the border between Liguria and Piedmont in 2022 gradually spread to the wild boar populations residing in these two regions; later, the disease also reached the Lombardy Region, where the first case was reported on 23 June 2023. The detection of ASF in Lombardy raised deep concern due to the region’s prominent position as the largest producer of pork products and derivatives in Italy, and one of the leading producers globally. Pig farming is a key pillar of Italian livestock farming, which is strong in many regional productions of the highest quality. In Lombardy, pig farms alone generate a value exceeding three billion euros, representing nearly 20% of the Italian livestock sector. Additionally, the cured meats industry accounts for over eight billion euros, constituting approximately 5.6% of the national food sector. Notably, Lombardy boasts several Protected Geographical Indication (PGI) cured meats, such as Prosciutto e Coppa di Parma, Cotechino di Modena, Mortadella Bologna, Salame Cremona, Salame d’oca di Mortara, and Zampone Modena, alongside numerous Protected Designation of Origin (PDO) products, including Salame Brianza, Salame di Varzi, and Salamini Italiani alla Cacciatora. To contain the spread of ASFV and safeguard the Italian pork meat industry, while also maintaining pig meat production for export, authorities swiftly defined restricted zones I (RZI) and II (RZII). Moreover, all private swine farms in Pavia province were closed. In RZII, activities such as dog training and hunting, mushroom picking, and outdoor recreational gatherings involving over 20 individuals were prohibited. Active research of carcasses was implemented in restricted zones; the territory was divided into 1 × 1 km grids and passive surveillance of carcasses was performed once per seven or fifteen days in restricted zone II or I, respectively. Given the heightened risk of ASFV introduction into domestic pig farms within the region, farms unable to meet enhanced biosecurity standards outlined in Reg. 594/2023 [27] were encouraged to cease operations, with preventive depopulation implemented in three farms (totalling 95 pigs) in RZII.

Furthermore, farm biosecurity controls were strengthened following the implementation of Regional Decree n. 2086 of 15 February 2023 [31], which mandated inspections of at least 50% of intensive farms housing >300 pigs, as well as all semi-free-range farms. Notwithstanding the prompt implementation of containment measures to stop ASFV spread, including the expansion of restricted zones, on 18 August 2023, the first case of ASFV positivity in a domestic pig was promptly reported in a swine-fattening farm with 166 animals in the Municipality of Montebello della Battaglia (Pavia Province). This area had previously identified an ASFV-positive wild boar two days prior. The epidemiological investigation suggested potential ASFV introduction via indirect contact with wild boars (human factor) and identified management gaps relating to cleaning and disinfection procedures, management of the clean area/contaminated area, and vehicle movements. However, the absence of animal movements and the farmer’s prompt reporting of the disease facilitated the quick containment and eradication of the outbreak, mitigating the risk of secondary outbreak-related cases. This first domestic outbreak heightened awareness of ASF in the region.

### 2.2. ASFV Infection Cluster in the Municipality of Zinasco

On 24 August 2023, an official veterinarian took the initiative to anticipate the scheduled inspection of biosecurity measure implementation at a swine-fattening farm of 1000 pigs in the Municipality of Zinasco (an ASFV-free area around 20 km from Montebello della Battaglia). This decision was prompted by an unusual flow of pigs to the slaughterhouse reported by Italian information systems. During the inspection, suspicions arose regarding an unreported significant mortality among the pigs, evident from the discrepancy between the number of present animals and the count of living and deceased pigs. Specifically, among the 49 pigs present, 26 were alive, while 23 were found dead. Tissue samples were collected from deceased animals for genotyping. Simultaneously, a rapid alert was issued to the Veterinary Service, leading to an immediate halt in the movement of live animals. Additionally, all slaughterhouses where the pigs had been dispatched were alerted.

On 28 August, the outbreak was confirmed and reported to the authorities. Subsequently, restricted zone III was implemented, and the Veterinary Services of Pavia, Regional Veterinary Public Health, and IZSLER conducted an ad hoc inspection aimed to carry out the epidemiological investigation. Upon inspection, only three living pigs were found on the farm, further underscoring the severity and impact of the disease.

#### The Epidemiological Investigation of the First Outbreak in Zinasco

The epidemiological observation of the IZSLER collected any data valid to reconstruct the farm history, aiming to identify the most likely origin of the contagion (trace-back) and the potential spread of the disease to other animals (trace-forward). This comprehensive data collection included farm details (location and farm owner), animal data (population size, production type), farm network characteristics (number and type of relationships with other farms), animal movements, external visits to the farm (veterinarians, breeders, salesmen), and clinical evaluation (number of dead and symptomatic pigs, date of disease suspicion, date of first symptom identification, type of symptoms, number of pigs serologically and virologically tested, number of pigs detected as ASFV-positive and ASFV-antibody-positive) [32]. 

The farm operated under an agistment contract. One thousand pigs were introduced to the farm in the previous months: five hundred pigs from a farm located in Brescia (Lombardy region) on 16 February 2023, and the other 500 animals from a farm located in Perugia (Umbria region) on 15 March 2023. The farmer (stockbreeders) first observed clinical signs in early August, characterized by recumbency and respiratory distress, attributing them initially to heatstroke due to high summer temperatures. The abnormality was communicated to the animal owner (animal bailers) and the farm veterinarian, who recommended a pharmacological treatment. The epidemiological investigation reconstructed a progressive increase in mortality. Despite experiencing approximately 350 pig deaths in the first weeks of the month, the increased mortality was not reported to the local Veterinary Service as required by regulations. This failure delayed diagnostic investigations and hindered the adoption of timely control and prevention measures. Concurrently, 564 pigs were slaughtered in three different slaughterhouses located in the Lombardy, Emilia-Romagna, and Veneto regions, starting from the fourth of August.

To establish the tracing window, August 1st was designated as day 0, when clinical signs were initially observed on the farm. This date was determined based on historical data and laboratory findings. Indeed, on the fourth of August, the farm was routinely checked for Aujeszky’s disease; 59 blood samples were collected in the slaughterhouse and sent to the local public veterinary laboratory (IZSLER), where they were analysed and stored. When the outbreak occurred, these samples were recovered and tested for ASF. Serological and virological investigations showed no ASFV positivity, suggesting the absence of infection at the sampling time (excluding a threshold prevalence > 5%). These findings were compatible with the recent introduction of the ASFV, aligning with the data collected during the epidemiological investigation. By subtracting the ASFV maximum incubation period (fifteen days, per WOAH (ex OIE) guidelines [33]) from day 0, the trace-forward activity commenced on 15 July.

Concurrently, investigations were conducted to identify infection entry points through animal/human/vehicle movements and direct/indirect contact with wild boars. The last animal introduction into the farm dated back to March 2023; therefore, this possible source of infection was considered unlikely. Moreover, there was no evidence of potentially contaminated humans entering the farm, no indirect contact with infected wild boars was suspected, and no animals were introduced to the farm during the trace window. On the other hand, the farm’s agistment contract facilitated dense vehicle-related contacts with numerous farms throughout Italy. Although the herd register revealed no vehicles with significant risk contacts, this route was considered the most probable entry point for ASFV. 

Protection and surveillance zones were established and active surveillance on pig farms, along with increased farmer and private veterinarian awareness within the province, were enhanced, which led to the identification of seven additional related outbreaks in the following weeks. In a 3.4 km^2^ area in the Municipality of Zinasco, four outbreaks in domestic pigs (three clinically suspected and one detected through laboratory investigations) were identified. One of them occurred in a “sanctuary” where pigs were cared for as pets. Two other outbreaks were detected in areas close to the surveillance zone in the Municipalities of Dorno and Sommo, respectively. The last outbreak of the Zinasco infection cluster was identified in the Municipality of Pieve del Cairo on 27 September 2023 (Figure 1).

### 2.3. Measures Applied to Stop ASFV Spread in Domestic Pigs in Lombardy Region 

The epidemiological investigation showed several links among the outbreaks, summarised in Table 1.

Among the infected pig farms, only two were involved in the movement of potentially infected animals: the location of the initial outbreak in Zinasco and the location of the final outbreak in Pieve del Cairo. Specifically, the epidemiological investigation conducted at the latter farm revealed the movement of 2800 pigs to a swine-fattening farm in Mantova, a province in the southernmost part of Lombardy, bordering the Emilia-Romagna region. Importantly, this movement of animals did not lead to the transmission of ASFV.

#### 2.3.1. Outbreak Eradication and Preventive Culling to Halt ASFV Spreading

Reported outbreaks were promptly eradicated by culling and safe carcasses disposal via rendering, except for at the largest farms, which required the activation of specific agreements, and at the sanctuary, where ethical concerns raised by animal rights activists necessitated additional considerations (Table 1).

Moreover, thorough epidemiological investigations conducted of each infected farm enabled the tracing of movements of people and vehicles at risk, such as trucks transporting feed, animals, and carcasses. One hundred twenty-seven pig farms were identified as having a moderate risk of ASFV introduction and were promptly placed under restriction by the Competent Authority (CA). Movements of animals were prohibited, and regular weekly clinical visits and sampling of animal tissues from dead animals for real-time PCR analysis were carried out, all of which yielded favourable outcomes.

Additionally, epidemiological investigations identified nine additional pig farms at high risk due to their proximity (within 10,294 m, corresponding to double the standard deviation among the outbreaks) to the centre of the Zinasco infection cluster. Six other farms were identified with strong epidemiological links, either through common ownership, belonging to the same pig production chain, or through the introduction of potentially infected animals. The CA implemented preventive culling measures on these swine farms based on the risk assessment. Table 2 provides details on the preventive culling and culling performed due to epidemiological links with confirmed outbreaks.

Regarding the culling of animals, the Lombardy region initially had an agreement with an Italian company specializing in animal killing through electrocution. However, the company struggled to meet the escalating demands prompted by the emergency epidemiological situation. Consequently, a further agreement was established with a Dutch company to swiftly eradicate all outbreaks and halt the spread of ASFV. The Dutch company employed gas chambers filled with carbon dioxide, a method approved by the expert group in animal welfare affiliated with the National Reference Laboratory for Animal Welfare of IZSLER. Conversely, the culling of pigs in the sanctuary presented a unique challenge. Here, an anaesthetic and euthanasia protocol was developed and applied thanks to academic support. 

Overall, approximately 4.5 million euros were allocated to ensure a swift and humane culling process in accordance with the provisions outlined in Council Regulation (EC) 1099/2009 [34] concerning the protection of animals at the time of killing and broader animal welfare principles. 

#### 2.3.2. Measures Applied to Restriction Zones

A protection zone extending 10 kilometres and a surveillance zone covering the rest of the Pavia province were established around the outbreaks in domestic pigs according to the Commission Implementing Decision (EU) 2023/1684 [35]. At the outset of September, the protection zone contained 18 swine farms housing 15,000 animals, with one farm accounting for 11,842 pigs. Meanwhile, the surveillance zone encompassed 79 farms with 198,869 animals. To mitigate the risk of disease spread and safeguard the Italian pig population, all farms within the protection and surveillance zones underwent official controls. Clinical examinations and assessments of biosecurity measures were conducted at all swine farms (97 in total). Additionally, in 71 out of 97 pig farms, 442 samplings were performed for diagnostic investigations. Controls yielded predominantly favourable results, except for five farms found to be non-compliant with biosecurity regulations. In both the protection and surveillance zones, restrictions were imposed on the movements of pigs (between farms or toward slaughterhouses), carcasses, and manure obtained from porcine animals, with specific exceptions outlined in Delegated Regulation (EU) 2020/687 [36]. No derogations were granted for pigs coming from the protection zone. Furthermore, derogations were only permitted under certain conditions:The destination farm was located within the province of Pavia.The slaughterhouse was situated within the Lombardy region.Dedicated transport vehicles were utilized for the operation.Vehicles underwent thorough washing and disinfection with ASFV-approved disinfectants after unloading.Veterinarians and technicians from the Pavia province were authorized to operate in other provinces or regions only after a minimum “inactivity” period of 5 days.

Additionally, housing of pigs coming from outside the province was prohibited within the protection and surveillance zones.

A designated slaughterhouse was identified according to Commission Implementing Regulation (EU) 2023/594 [30] to allow the slaughter of pigs coming from RZII.

#### 2.3.3. Measures Applied Outside the Restriction Zones in the Lombardy Region

Several provisions were promptly issued across the remaining regional territory to establish an early detection system consistent with the epidemiological situation and to mitigate risks. Specifically, an extraordinary procedure governing the movement of pigs towards other establishments or slaughterhouses was defined. Animal movements were permitted provided that: (1) a clinical examination and assessment of mortality trends were conducted twenty-four hours before the first load and subsequently repeated every 72 h and (2) molecular tests on spleen samples from carcasses of animals that died recently, or, in the absence of such samples, blood sampling from animals exhibiting diminished vitality, carried out 72 h before the first load, and then reiterated every 72 h, showed favourable results. 

Consequently, all farms involved in animal movements, whether for breeding purposes or slaughter, underwent clinical examinations and mortality trend assessments, all of which yielded favourable outcomes. Nearly 9745 inspections have been conducted. Additionally, diagnostic investigations were carried out in 1566 of these farms. Between 28 August and 26 October, 9140 real-time PCR tests (8561 on spleen and 579 on blood samples, respectively) were performed. Approximately 220 thousand euros have been allocated for laboratory tests. Furthermore, CA made the validation of animal movement forms mandatory in all regional territories. Moreover, measures included the following:a prohibition on moving pigs for participation in fairs, markets, and exhibitions;a requirement for rendering companies to promptly communicate any observed increase in the frequency of carcass disposal from pig farms;a straightening of CA-led inspections of pig farms to ensure strict adherence to biosecurity measures (over the course of approximately four months, 52% (1081/2060) of swine farms were inspected in the Lombardy region (Table 3)).

### 2.4. Measures Applied to Stop ASFV Spread by Meat and Animal By-Products (ABP)

Based on the findings of the epidemiological investigation, pigs infected or potentially infected from the first outbreak in Zinasco were moved to three slaughterhouses located in Lombardy (Mantova province), Emilia-Romagna (Modena province), and Veneto (Verona province) regions during the tracing window. CCU and GOE implemented the following measures to prevent the spread of ASFV through meat and ABP:distribution lists of meat and ABP produced from the infected or potentially infected pigs slaughtered in Mantova (240 animals), Modena (250 animals), and Verona (74 animals) were shared with Veterinary Services and the Ministry of Health.an administrative assistance notification, categorized as “non-compliance and potential risk for animal health” was registered on the Rapid Alert System for Food and Feed (iRASFF) (https://webgate.ec.europa.eu/irasff, accessed on 22 April 2024). This aimed to initiate all necessary actions to prevent ASFV spread, adhering to the principle of maximum precaution, including the withdrawal and recall of meat and ABP at the national and Member States level. These included the following:
meat and ABP derived from animals reared in infected farms;meat and ABP derived from animals reared in farms that had contact with animals from infected farms during slaughter on the same days and without carcasses segregation;meat and ABP derived from animals from lairages where animals reared in the infected farm were present, provided the contact was sufficient to induce viremia before slaughter (more than four days, the minimum incubation time for ASFV [37]);meat and ABP that entered into contact (e.g., mixing or grinding) with those mentioned above during any movements.


For this purpose, the local CA implemented the following:withdrawal and destruction of meat and ABP falling within the cases mentioned in point 1, classified as “category two” according to Reg. (EC) 1069/2009 [38];withdrawal and blocking of meat and products falling within the cases mentioned in points 2, 3, and 4. A specific risk assessment, supported by regional epidemiological observatories, was carried out to establish the fate of confiscated meat and ABP.

Products and semi-finished products subjected to ASFV inactivation processes, according to Annex VII of Delegated Regulation (EU) 2020/687 [36], were not subject to tracing and recall, except as provided in the cases referred to in point 1.

### 2.5. Consequences of Zinasco Cluster

Despite the extensive efforts made by institutions to mitigate the consequences of the Zinasco cluster of infection, certain events proved unavoidable.

When the first ASF outbreak in Zinasco was identified, ASFV in wild boars was confined to the territory of Piedmont and Liguria with only isolated cases reported along the Lombardy–Piedmont boundary. On 23 August 2023, an infected wild boar was found in the municipality of Montesegale (Pavia province) (Figure 2A), approximately seven kilometres from the Piedmont boundary. Given the solid epidemiological link and the evolving ASF situation in the domestic sector in Lombardy, mandatory ASFV testing of all wild boars hunted in the Pavia province was instituted. Two ASFV-positive wild boars were identified in the Ticino Valley Natural Park on 5 and 6 October (Figure 2B), a national and regional protected green area characterized by water courses, conifer, moorland, and wetlands that had been ASF-free in preceding months and situated approximately 30 kilometres from the nearest wild boar cluster. Subsequently, thanks to the strengthened passive surveillance implemented by the Ticino Park Authority, two additional wild boars were found in the same area on 11 and 12 October 2023. Presently (20 January 2024), approximately twenty cases of ASF in wild boars have been reported in Ticino Park, delineating an isolated cluster of infection in wild boars likely stemming from a domestic cluster of infection in Zinasco (Figure 2B).

Following the spread of ASFV within the wild boar population, the infected area of Lombardy expanded and was reclassified into restricted zones I and II of northern Italy. On the other hand, a reassessment of regionalization was undertaken due to the positive trend in the epidemiological situation among domestic pigs, marked by the absence of new outbreaks for over three months. On 25 January 2024, in accordance with the Reg. (UE) 2024/413, restricted zone III in Lombardy was downgraded to restricted zones I and II [39]. 

On 20 September 2023, the first outbreak of ASFV genotype II was reported at a small farm in the municipality of Dorgali (Nuoro province, Sardinia island) [40]. The epidemiological investigation revealed that the virus had crossed the sea and infected the pigs through the consumption of food waste. Tracing the meat and meat products suspected of causing the ASFV genotype II outbreak identified an indirect link with the first outbreak in Zinasco [40]. Indeed, as reported in a previous study, on 23 August, infected meat products were sold from the wholesalers of Emilia-Romagna to three Sardinian retailers, located in the northwest, northeast, and south of the island. The meat products arrived on the island on 24 August and were recalled on 29 September. However, a part of the meat and meat products escaped from the recall process. They were acquired by a butcher who fed his pigs with food waste (likely including the infected meat) on 26 August. Exactly 15 days later (10 September), the first three pigs died and, subsequently, another three died on 19 September [40]. 

## 3. Discussion

Thanks to the rigorous surveillance measures and eradication actions put in place, ASFV was eradicated in domestic pigs in Lombardy. Nevertheless, the spread of the ASFV in wild boars in northwest Italy has proven to be relentless, with affected areas continuing to expand. The transmission of the virus amongst wild boar populations poses a significant threat to ASF-free areas. When the disease crossed the regional borders of Piedmont to Lombardy, the risk of ASFV spreading to domestic pig farms increased substantially. As in other Italian regions [9] and in other countries [19,41], when the ASFV spreads and becomes endemic in wild boar populations, it eventually also affects domestic pigs. While the first case of ASF in domestic pigs in Lombardy (in the Municipalities of Montebello della Battaglia) was promptly reported and eradicated, the situation in Zinasco unfolded unpredictably.

Between 18 August and 27 September 2023, nine ASFV outbreaks occurred in Lombardy, all within the province of Pavia. It is notable that, except for the first outbreak in Montebello della Battaglia, which occurred in an area already under restrictions due to the presence of the disease in wild boars, the remaining outbreaks transpired in territories where surveillance of wild boars had not uncovered any indication of the disease’s presence.

The first outbreak in Zinasco was pivotal for the subsequent outbreaks of the area. It was detected during an official control activity planned by the Veterinary Services of Pavia as a follow-up to a control carried out in June, anticipated due to anomalous animal movements. The delayed early detection of ASF and the failure to implement adequate control and prevention measures allowed the virus to spread further in the environment for approximately three weeks, resulting in a high viral load in the municipality of Zinasco and an increased risk of introducing the disease to nearby farms. This, coupled with other risk factors, such as shared supply chains and animal ownership (animal bailers), contributed to the occurrence of the following outbreaks. In contrast, in the other outbreaks, the cooperation and awareness of farmers, along with coordinated efforts across the chain of command, facilitated early detection of the disease and swift implementation of effective control measures, safeguarding the pig breeding industry in Lombardy and beyond.

The implementation of various measures to combat the ASF outbreak has been a remarkable challenge. Regulations have been rigorously enforced in restriction zones, with authorities swiftly eradicating outbreaks and implementing preventive culling measures. Conversely, exceptional procedures have been implemented to move pigs outside these zones in Lombardy. In Lombardy alone, over 1500 farms have been checked, covering approximately 75% of the intensive pig herd, all within a month and a half. Nearly 9745 inspections have been conducted to monitor the mortality trend and the health status of pigs. At the same time, 9140 real real-time PCR tests have been performed to facilitate early disease detection and enable the movement of animals to other farms or slaughterhouses. Rendering companies have been urged to promptly report any increase in the frequency of carcass disposal from pig farms. Biosecurity measures applied to pig farms have been subject to enhanced scrutiny, and measures have been taken to prevent the spread of ASF through meat and ABP. These efforts have required significant dedication and resources, both in terms of finances and personnel. Approximately 4.5 million euros have been allocated for managing culling efforts, with an additional 13.5 million euros earmarked for compensating farmers. The substantial financial investment underscores the Italian government’s commitment to addressing this emergency.

The culling conducted within restricted areas of Zinasco, and the involvement of the “sanctuary” attracted significant media attention. Activist groups mobilized, garnering support from a broad swath of public opinion and even resorting to obstructionist tactics against authorities. This scenario created an emergency within an emergency. Heightened awareness of ethical concerns in farming practices and killing and animal welfare, along with the rapid dissemination of information by the media, have empowered individuals to exert influence over official decisions. In this case, animal rights activists deemed the mass culling of pigs unacceptable despite the severe, deadly, and painful nature of ASF and the stringent laws in place to protect animal welfare. They questioned the necessity of sacrificing thousands of animals for health reasons, highlighting a tension between public perception, ethical considerations, and disease control measures.

## 4. Conclusions

The outbreak of infection among domestic pigs in Zinasco posed a severe threat to the Lombardy region, requiring a prompt response in accordance with national and European regulations. Measures outlined in Regulation (EU) 2020/687 were extended to the entire province of Pavia, as per Commission Decision 2023/1684 dated 31 August 2023 [35]. Moreover, ad hoc provisions were implemented to address the extraordinary emergency, adhering to the principle of maximum precaution, which helped mitigate the risk of ASFV spreading and provided clarity on the epidemiological situation.

However, these restrictions also raised significant concerns. As per Annex VII of Regulation (EU) 2020/687 [36], the obligation to implement risk reduction methods in surveillance zones restricted meat usage. Given that most pig farms in Lombardy are specialized in heavy pig production for cured meat, this limitation led to overcrowding in pig farms and, subsequently, the need to cull animals that could not be relocated, resulting in both economic and ethical implications.

Despite these challenges, Italian authorities responded promptly and effectively to contain the infection, confining it primarily to the Pavia area. Only one case was reported outside the Lombardy region, detected in a domestic pig farm in Dorgali (Nuoro province, Sardinia), showing an epidemiological link with the Zinasco cluster [40]. The joint efforts implemented in Italy facilitated the eradication of ASF in domestic pigs in just over 1.5 months, representing an example of effective and timely cooperation to mitigate the spread of the infection. As of approximately four months after the initial outbreak, ongoing control activities indicate a stable and controlled epidemiological situation, thanks to heightened awareness across the entire pig production chain regarding passive surveillance and the importance of implementing biosecurity measures.

The Italian experience underscores the importance of considering the societal impact of sanitary measures outlined in current legislation. In the Italian scenario, two different approaches were employed to cull animals in ASFV-infected outbreaks: one for the livestock, in accordance with legislation, and another for animals kept in the sanctuaries, taking into consideration people’s sentiments towards animals considered as pets. Despite efforts to respect people’s ethical concerns, ethical issues remain a significant challenge to address. Balancing the people’s sentiments regarding ethical concerns with the risk of ASF affecting densely populated livestock districts will be challenging, necessitating proactive prevention efforts to curb virus transmission. Innovative approaches and increased resources will be essential for eradicating the virus in wild boars and preventing its spread to domestic pigs.

## Figures and Tables

**Figure 1 animals-14-01295-f001:**
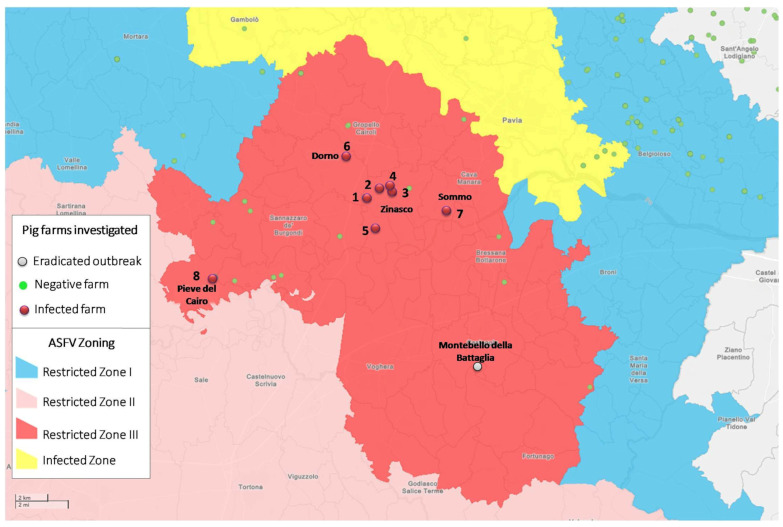
Mapping of epidemiologically linked outbreaks stemming from the initial outbreak in the Municipality of Zinasco. The outbreaks are numbered chronologically; number one represents the initial outbreak in Zinasco, which occurred at the fattening farm housing approximately 1000 pigs. The first outbreak at a domestic pig farm in Lombardy (Montebello della Battaglia) that has already been eradicated is indicated in grey.

**Figure 2 animals-14-01295-f002:**
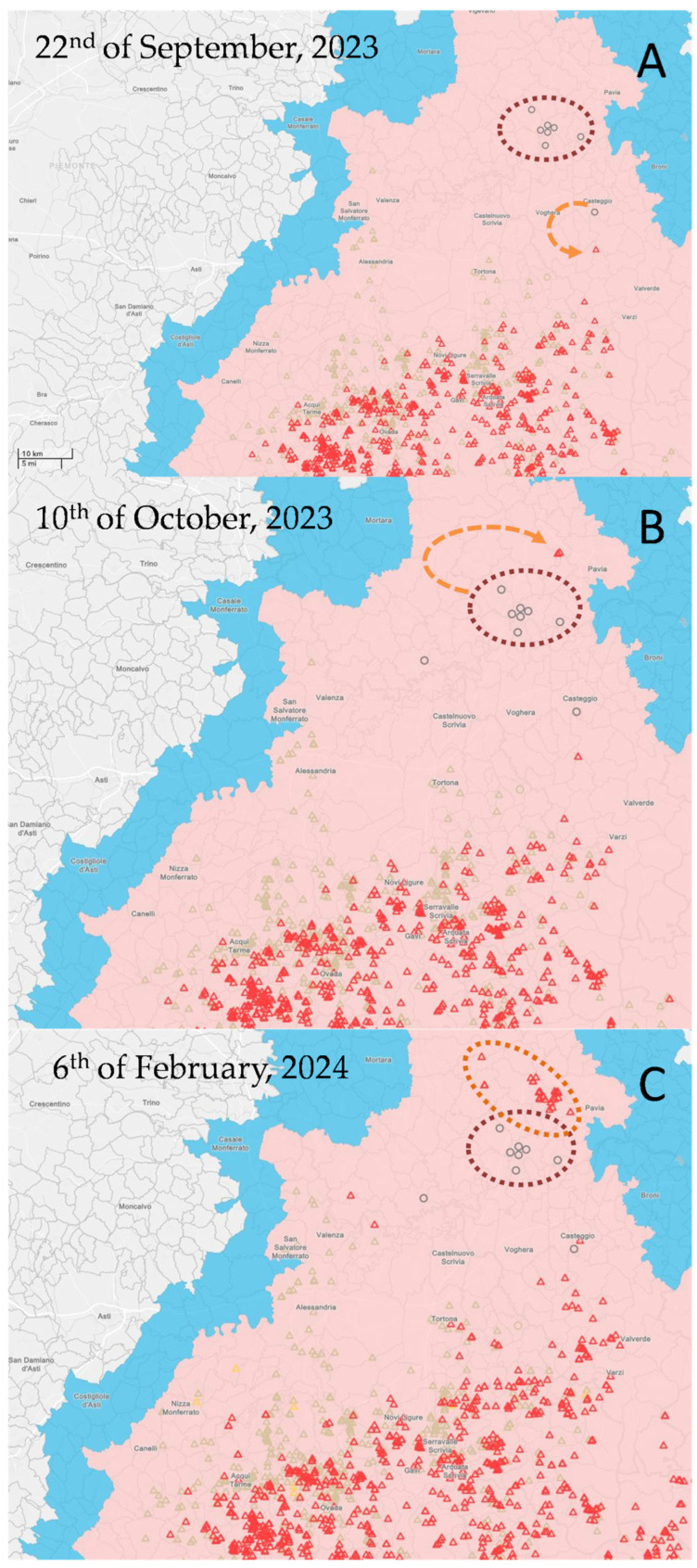
African swine fever epidemiological evolution in wild boars in Lombardy region. (**A**) Epidemiological scenario updated to 22 September 2023: all the infected domestic outbreaks in Pavia province were eradicated; the first ASFV-positive boar epidemiologically linked to the first domestic outbreak in Lombardy (Montebello della Battaglia) was reported. The Zinasco infection cluster is surrounded by a red dotted line; the orange dotted arrow describes the epidemiological link between the domestic outbreak in Montebello della Battaglia and the nearby first-infected wild boar. (**B**) Epidemiological scenario updated to 10 October 2023: the first two ASFV-positive wild boars in Ticino Park linked to the Zinasco cluster of infection were reported. The Zinasco infection cluster is surrounded by a red dotted line; the orange dotted arrow describes the epidemiological link between the Zinasco infection cluster and the first two ASFV-positive boars found in Ticino Park. (**C**) Epidemiological scenario updated to 6 February 2024: a cluster of infection in wild boars linked to the Zinasco infection cluster is spreading in the Ticino Park. The Zinasco infection cluster is surrounded by a red dotted line; the orange dotted line surrounds the ASF infection in wild boars epidemiologically linked to the Zinasco infection cluster. Blue and pink areas represent RZ I and RZ II, respectively. Grey circles are eradicated domestic outbreaks; red triangle are infected wild boars; light green triangles are uninfected wild boars.

**Table 1 animals-14-01295-t001:** List of the outbreaks in the Zinasco infection cluster and epidemiological links among them.

Outbreaks #	Province	Municipality	Category	Outbreak Type	Data Reported	Data of Epidemiological Investigation	Data of Outbreak Eradication	Number of Animals			
								Before the Outbreak	On the Farm	Dead	Culled
1	Pavia	Zinasco	Fattening farm	Clinical outbreak	28/08/2023	29/08/2023	29/08/2023	1000	26	23	3
2 *^,†,‡,•^	Pavia	Zinasco	Fattening farm	Clinical outbreak	28/08/2023	31/08/2023	07/09/2023	2237	2206	31	2206
3 *^,†,‡,•^	Pavia	Zinasco	Fattening farm	Diagnostic positivity	30/08/2023	31/08/2023	06/09/2023	7428	7428	1	7427
4 *	Pavia	Zinasco	Fattening farm	Clinical outbreak	31/08/2023	03/09/2023	04/09/2023	4	4	1	3
5 *	Pavia	Zinasco	Sanctuary (pigs kept as pets)	Clinical outbreak	04/09/2023	04/09/2023	22/09/2023	40	40	2	38
6 *^,•^	Pavia	Dorno	Fattening farm	Clinical outbreak	04/09/2023	06/09/2023	10/09/2023	1191	1191	2	1189
7 ^†,•^	Pavia	Sommo	Fattening farm	Clinical outbreak	07/09/2023	08/09/2023	13/09/2023	1850	1850	10	1840
8 ^†^	Pavia	Pieve del Cairo	Breeding farm	Clinical outbreak	27/09/2023	28/09/2023	07/10/2023	6864	4068	23	4056
	Total animals culled										19,644

* = near to the first outbreak; ^†^ = belonging to the same pig production chain; ^‡^ = same owner; ^•^ = same operator in charge of the carcasses’ disposal.

**Table 2 animals-14-01295-t002:** Preventive culling carried out in the Lombardy region.

**PREVENTIVE CULLING DUE TO CLOSE PROXIMITY TO AN OUTBREAK**				
**Pig Farming**	**Province**	**Category**	**Number of Animals**	**Culling Date**
1	Pavia	Fattening pigs	29	08/09/2023
2	Pavia	Fattening pigs	12	08/09/2023
3	Pavia	Fattening pigs	7	08/09/2023
4	Pavia	Fattening pigs	3	08/09/2023
5	Pavia	Fattening pigs	1	08/09/2023
6	Pavia	Fattening pigs	250	12/09/2023
7	Pavia	Fattening pigs	3743	14/09/2023
8	Pavia	Fattening pigs	90	15/09/2023
9	Pavia	Fattening pigs	8770	21/09/2023
Total animals killed for preventive culling due to close proximity to an outbreak			12,905	
**PREVENTIVE CULLING DUE TO AN EPIDEMIOLOGICAL LINK WITH AN CONFIRMED OUTBREAK**				
**Pig farming**	**Province**	**Category**	**Number of animals**	**Culling date**
1	Pavia	Weaner	1905	06/09/2023
2	Pavia	Fattening pigs	1753	07/09/2023
3	Pavia	Fattening pigs	3494	11/09/2023
4	Pavia	Stores	720	07/09/2023
5	Pavia	Weaner	1200	07/09/2023
6	Mantova	Piglets and weaners	4791	01/10/2023
Total animals killed for preventive culling due to the epidemiological link			13,863	
Total animals culled			46,578	

**Table 3 animals-14-01295-t003:** Inspection of the biosecurity measures applied to farms in the Lombardy region from 01/07/2022 to 20/10/2023. A single farm could be checked more than once.

Province	Number of Controlled Farms	Number of Inspections	Number of Inspections with Favourable Outcome	Number of Inspections with Unfavourable Outcome
Bergamo	81	116	85	31
Brescia	260	345	258	87
Brianza	25	42	27	15
Insubria	43	54	13	41
Milano	169	266	122	144
Montagna	27	43	22	21
Pavia	87	130	92	38
Val Padana	389	558	422	136
Total	1081	1554	1041	513

## Data Availability

No new data were created or analysed in this study. Data are contained within the article.

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
