# Peer review of "Strategic Challenges to the Eradication of African Swine Fever Genotype II in Domestic Pigs in North Italy"

_animals, 2024, doi:10.3390/ani14091295_

Round 1

Reviewer 1 Report

Comments and Suggestions for Authors

The emergence of African Swine Fever Virus (ASFV) genotype II into Georgia in 2007 has exemplified the challenges related to controlling this disease. Pavone et al. describe a recent outbreak of ASFV in domestic pigs within the Zinasco municipality of Lombardy, Italy. The beginnings of the outbreak this article describes were traced to a wild board in January 2022. Subsequent detection of ASFV-positive wild boars in the middle of 2023 prompted the implementation of various control measures to halt the spread of the virus into Italian livestock (pig) farms. The authors clearly describe the chronology of events, provide details on the various control measures implemented (biosecurity enforcement, movement restrictions, and culling), and highlight the challenges faced. Overall, the rapid responses and coordinated efforts to limit the spread of ASFV within Lombardy and prevent spill-over into adjacent provinces described in this report may serve as a blueprint for handling future outbreaks in livestock.

Areas of improvement and reader clarification:

-      Were the authors able to identify potential sources for the ASFV introduction in the farm in Zinasco?

-      Can the authors provide information on any investigation(s) performed into the delayed reporting of the first outbreak?

-      The text mentions an agistment contract for the pigs in Zinasco. Were there any biosecurity measures or policies in place prior to the discussed outbreak that should have prevented disease introduction?

-      With respect to the ASFV genotype II outbreak in Dorgali, has any investigation into the source of the contaminated ABP been identified? The date of this outbreak would suggest control efforts limiting the spread of ASFV out of Lombardy were already in place, leaving the reader wondering whether certain policies were not followed or if a potential alternative source for the virus exists.

-      Figure 1: Consider adding location symbols of the three slaughterhouses used by the farm traced to the first outbreak in Zinasco.

-      It is unclear how the text between lines 261-271 are relevant to the overall conclusions of this article.

-      Figure 2:

o   When is the start date for surveillance?

o   In the figure legend, consider changing “Infected wild boar” to “ASFV-positive boar”, and “Not Infected wild boar” to “ASFV-negative boar”.

o   Consider inserting the date within each of the three panels to help the reader understand the timeline.

o   Highlight the new positive/negative boars from A-to-B and B-to-C. For example, make the positive/negative symbols that are not in A, but are in B, a more saturated color. This will help the reader visualize the outbreak spread.

o   Consider removing the ASFV zoning from the legend and describing it within the figure legend.

-      Table 1:

o   Change “Outbreaks” header to “Outbreak #” or “Outbreak number” or “Outbreak”.

o   Change “Data of notified” header to “Date notified”

Comments on the Quality of English Language

The following (line-by-line) were areas of text that need clarification or editing. I have provided potential alternatives.

113: Moreover, all private swine farms in Pavia province were closed. 

116-120: Active research of carcasses was implemented in restricted zones; the territory was divided into 1x1 km grids and passive surveillance of carcasses were performed once per seven or fifteen days in restricted zone II or I, respectively.

148-149: Tisue samples were collected from deceased animals for genotyping.

153: Spelling error; implemented

155: aimed to carry out the epidemiological investigation.

159: The epidemiological observation

233-236: Specifically, the epidemiological investigation conducted in the latter farm revealed the movement of 2,800 pigs to a swine fattening farm in Mantova, a province in the southernmost part of Lombardy. Importantly, this movement of animals did not lead to the transmission of ASFV.

240-241: ..., and for the sanctuary where ethical concerns raised by animal rights activists necessitated additional considerations.

318-319: ..., all of which yielded favorable outcomes.

378-383: Through the implementation of passive surveillance and collaboration of citizens, Two ASFV-positive wild boars... from the nearest wild boar cluster.

417-419: Tracing the meat and meat products suspected of causing the ASFV genotype II outbreak identified an indirect link with the first outbreak in Zinasco. 

Author Response

Reviewer 1

The emergence of African Swine Fever Virus (ASFV) genotype II into Georgia in 2007 has exemplified the challenges related to controlling this disease. Pavone et al. describe a recent outbreak of ASFV in domestic pigs within the Zinasco municipality of Lombardy, Italy. The beginnings of the outbreak this article describes were traced to a wild board in January 2022. Subsequent detection of ASFV-positive wild boars in the middle of 2023 prompted the implementation of various control measures to halt the spread of the virus into Italian livestock (pig) farms. The authors clearly describe the chronology of events, provide details on the various control measures implemented (biosecurity enforcement, movement restrictions, and culling), and highlight the challenges faced. Overall, the rapid responses and coordinated efforts to limit the spread of ASFV within Lombardy and prevent spill-over into adjacent provinces described in this report may serve as a blueprint for handling future outbreaks in livestock.

Dear Reviewer,

We are pleased that the manuscript was appreciated as a useful contribution for handling future ASFV outbreaks in livestock. We sincerely appreciate your assistance in significantly improving the manuscript. We made the changes following the suggestions received to meet the requirements.

Thank you once again for your valuable feedback.

Areas of improvement and reader clarification:

-      Were the authors able to identify potential sources for the ASFV introduction in the farm in Zinasco?

As reported in lines 198-201, the potential source for the ASFV introduction in the first farm affected in Zinasco was identified in vehicles, as other routes were excluded through epidemiological investigation. This conclusion was drawn because there was no evidence of potentially contaminated humans entering the farm, no indirect contact with infected wild boars was suspected, and no animals were introduced to the farm during the trace window. However, it's important to note that absolute certainty was not achieved, as no significant risk contacts were detected among the vehicles and other infected farms. Additional information has been included in the text to further clarify our considerations regarding potential sources."

-      Can the authors provide information on any investigation(s) performed into the delayed reporting of the first outbreak?

The alleged negligence by the breeder in reporting the observed increase in mortality on the farm late is the subject of an ongoing investigation by the Public Prosecutor's Office of Pavia.

-      The text mentions an agistment contract for the pigs in Zinasco. Were there any biosecurity measures or policies in place prior to the discussed outbreak that should have prevented disease introduction?

                Thank you to the Reviewer for the question, as it allows us to incorporate further details into the article. Indeed, since 2022, specific biosecurity measures for swine farms have been mandatory in Italy, as outlined in the Decree of June 28, 2022, which establishes biosecurity requirements for swine farms. These measures were implemented to enhance and harmonize the biosecurity level of Italian pig farms. Furthermore, since 2023, reinforced biosecurity measures have been applied in infected zones or restricted zones I, II, and III, following the provisions outlined in Annex III of Commission Implementing Regulation (EU) 2023/594. This information has been included in the text, specifically in lines 75-79."

With respect to the ASFV genotype II outbreak in Dorgali, has any investigation into the source of the contaminated ABP been identified? The date of this outbreak would suggest control efforts limiting the spread of ASFV out of Lombardy were already in place, leaving the reader wondering whether certain policies were nt followed or if a potential alternative source for the virus exists.

 Thank you very much for your remark, as it allows us to provide a clearer explanation of the ASF case in Dorgali. As mentioned in the text, until August 28th, when the Regional Authorities confirmed and notified the outbreak and initiated culling, animals from the affected farm continued to be sent to the slaughterhouse. On August 23rd, meat products from the farm were sold by wholesalers in Emilia-Romagna to three Sardinian retailers, located in the northwest, northeast, and south of the island. These products arrived on the island on August 24th and were recalled on September 29th. However, a portion of the meat and meat products were not seized; instead, they were retained by the butchery owner who fed them to his pigs, possibly including infected meat, on August 26th. Exactly 15 days later, on September 10th, the first three pigs died, followed by another three on September 19th. This sequence of events was documented in the recent paper titled 'The Long-Jumping of African Swine Fever: First Genotype II Notified in Sardinia, Italy' by Silvia Dei Giudici."

 -      Figure 1: Consider adding location symbols of the three slaughterhouses used by the farm traced to the first outbreak in Zinasco.

Thank to the Reviewer for the suggestions. Given the geographical dispersion of the three slaughterhouses across different regions and their distance from the outbreak in Lombardy, it may not be practical to include them in Figure 1. Adding them could result in a cluttered and unclear representation of the data. It's important to prioritize clarity and readability in visualizations, so we will refrain from including them in this case. Moreover, an image representing the tracing among the first outbreak in Zinasco and the three slaughterhouses has been already published by others Italian collegues in the article titled “The Long-Jumping of African Swine Fever: First Genotype II Notified in Sardinia, Italy” (Silvia Dei Giudici et al., 2024). This article has been cited in our paper.

-      It is unclear how the text between lines 261-271 are relevant to the overall conclusions of this article.

The methods used to cull the animals in ASFV-infected outbreaks are described in lines 261-271. Two different approaches were employed: one for the livestock, in accordance with legislation, and another for animals kept in the sanctuaries, taking into consideration people's sentiments towards animals considered as pets. This description is useful in explaining that, despite efforts to respect people's ethical concerns, ethical issues remain a significant challenge to address. We have now added a sentence in the conclusion to clarify better the measures applied and described in the results.

   -      Figure 2:

o   When is the start date for surveillance?

In 2019, a surveillance plan was developed and implemented in Italy, starting from 2020.

o   In the figure legend, consider changing “Infected wild boar” to “ASFV-positive boar”, and “Not Infected wild boar” to “ASFV-negative boar”.

                Modified accordingly.

 o   Consider inserting the date within each of the three panels to help the reader understand the timeline.

                Modified accordingly.

 o   Highlight the new positive/negative boars from A-to-B and B-to-C. For example, make the positive/negative symbols that are not in A, but are in B, a more saturated color. This will help the reader visualize the outbreak spread.

Thank to the Reviewer for the suggestion. Given the reviewer's requirement to remove the ASF zoning from the legend (the suggestion has been received), the adding further colours to the map may generate potential disorientation of the reader. Additionally, since new cases are already effectively highlighted by arrows and dotted circles, we think that the emphasis on new cases is adequately conveyed.

 o   Consider removing the ASFV zoning from the legend and describing it within the figure legend.

                Modified accordingly.

 -      Table 1:

o   Change “Outbreaks” header to “Outbreak #” or “Outbreak number” or “Outbreak”.

                Modified accordingly.

o   Change “Data of notified” header to “Date notified”

                Modified accordingly.

Comments on the Quality of English Language

The following (line-by-line) were areas of text that need clarification or editing. I have provided potential alternatives.

113: Moreover, all private swine farms in Pavia province were closed. 

                Modified accordingly.

116-120: Active research of carcasses was implemented in restricted zones; the territory was divided into 1x1 km grids and passive surveillance of carcasses were performed once per seven or fifteen days in restricted zone II or I, respectively.

                Modified accordingly

148-149: Tisue samples were collected from deceased animals for genotyping.

                Modified accordingly

153: Spelling error; implemented

                 Thank to the Reviewer for the feedback. The text was implemented.

155: aimed to carry out the epidemiological investigation.

                Modified accordingly

159: The epidemiological observation

                Modified accordingly

233-236: Specifically, the epidemiological investigation conducted in the latter farm revealed the movement of 2,800 pigs to a swine fattening farm in Mantova, a province in the southernmost part of Lombardy. Importantly, this movement of animals did not lead to the transmission of ASFV.

                Modified accordingly

240-241: ..., and for the sanctuary where ethical concerns raised by animal rights activists necessitated additional considerations.

                Modified accordingly

318-319: ..., all of which yielded favorable outcomes.

                Modified accordingly

378-383: Through the implementation of passive surveillance and collaboration of citizens, Two ASFV-positive wild boars... from the nearest wild boar cluster.

                Modified accordingly

417-419: Tracing the meat and meat products suspected of causing the ASFV genotype II outbreak identified an indirect link with the first outbreak in Zinasco. 

                Modified accordingly

Reviewer 2 Report

Comments and Suggestions for Authors

The manuscript entitled " Strategic challenges to eradicate African Swine Fever genotype 2 II in domestic pigs in North Italy” contains some interesting information on ASFV however needs to be improved.

The article is interesting and describes the ASF status in Italy. The main problem of the article is the lack of comparative information assessing the characteristics of ASF in Italy. This will complicate the assessment and analysis of the information provided by the authors. Below are the main comments in this context.

In the introduction section, it is necessary to add the reasons why the disease has spread so successfully throughout the Eurasian region, and then compare it with the situation in Italy. The similarities and differences in the distribution of ASF need to be explained. Are there any specific features of the spread of ASF in Italy?

Clinical descriptions of ASF cases are not mentioned (whether manifestations of chronic or asymptomatic forms are possible). Analysis of clinical material is important because it can explain one of the possible mechanisms of virus transmission.

It is necessary to clarify what contacts between wild boars and domestic pigs could lead to the transmission of the disease. The sentence "The epidemiological investigation suggested potential ASFV introduction via indirect contact with wild boars…" is clearly insufficient to understand the overall state of virus transmission and requires additional details. What do the authors mean by "indirect contact"? Are there any reliably documented cases of virus transmission from wild boar to domestic pigs?

It is also necessary to significantly expand this part of the article by adding data on the spread of the virus in the environment (specific features in the region being described).

The safety of the virus in the environment when pigs are free-grazing, and in water if it does not undergo sanitary cleaning, also requires clarification.

There are no described cases of burial of pig bodies as a source of re-infection (and it is not clear whether such cases are not observed or are simply not described enough).

In other words, the article needs to pay more attention to possible routes of transmission of the virus in the region and their characteristics (if any).

Author Response

Reviewer 2

The manuscript entitled "Strategic challenges to eradicate African Swine Fever genotype 2 II in domestic pigs in North Italy” contains some interesting information on ASFV however needs to be improved. The article is interesting and describes the ASF status in Italy. The main problem of the article is the lack of comparative information assessing the characteristics of ASF in Italy. This will complicate the assessment and analysis of the information provided by the authors. Below are the main comments in this context.

Dear Reviewer,

We are pleased to hear that the manuscript is interesting and useful for handling future ASFV outbreaks in livestock. We sincerely appreciate your assistance in significantly improving the manuscript. We made the changes following the suggestions received to meet the requirements. Thank you once again for your valuable feedback.

In the introduction section, it is necessary to add the reasons why the disease has spread so successfully throughout the Eurasian region, and then compare it with the situation in Italy. The similarities and differences in the distribution of ASF need to be explained. Are there any specific features of the spread of ASF in Italy?

Thank to the Reviewer to give us the opportunity to add more information to the article. Additional details have been incorporated into the introduction as follows:

1) The epidemiological cycle involving the Eurasian wild boar (Sus scrofa), known as the wild boar–habitat cycle [5], represents a significant challenge in combating ASF in these territories. A growing population of wild boars, which serve as the main reservoir of ASFV in the environment, increases the chances of the virus spreading into new geographical areas and persisting long-term in the wild, thereby heightening the risk of its introduction into the domesticated swine population [25]. A combination of control measures, including fencing, sniper-guided shooting and trapping of wild boars, along with the crucial task of searching for and disposing of wild boar carcasses, tailored to the epidemiological situation, is currently considered effective for ASF control [26]. However, variations among countries in landscape, farming practices (free-range or semi-free-range versus intensive farming), and the level of biosecurity measures applied in swine farms may influence the management of the disease and the success of implemented actions.

2) In Italy, as in other European countries, wild boar populations have experienced significant expansion resulting in the colonization of new habitats such as agricultural lands, peri-urban, and urban areas [28]. The Italian Apennines showed a high wild boar population, ranging from four to eighteen animals for Km2 [28]. Specifically, the local wild boar density in Liguria and Piedmont was estimated in 60,000 and 85,000, respectively [9].

3) However, the unique landscape characterised by rocky terrain, rivers and bridges, encompassing urban areas and bordered by rural and forested areas, , coupled with the lengthy bureaucratic processes in Italy, impeded the construction of effective fences to halt the spread of ASF. On the other hand, a specific Decree detailing generic biosecurity measures for swine farms was implemented on June 28, 2022, to protect the pig sector [29]. Furthermore, reinforced biosecurity measures have been applied in infected zone or restricted zone I, II, and III following the provisions outlined in annex III of Commission Implementing Regulation (EU) 2023/594 [30].

Clinical descriptions of ASF cases are not mentioned (whether manifestations of chronic or asymptomatic forms are possible). Analysis of clinical material is important because it can explain one of the possible mechanisms of virus transmission.

More information on clinical signs observed in diseased animals has been added in par 2.2.1

It is necessary to clarify what contacts between wild boars and domestic pigs could lead to the transmission of the disease. The sentence "The epidemiological investigation suggested potential ASFV introduction via indirect contact with wild boars…" is clearly insufficient to understand the overall state of virus transmission and requires additional details. What do the authors mean by "indirect contact"? Are there any reliably documented cases of virus transmission from wild boar to domestic pigs?

The term “Indirect contact” has been now explained in the text. There were no reliably documented cases of virus transmission from wild boar to domestic pigs. However, as described in the article, the epidemiological investigation identified some gaps in cleaning and disinfection procedures, management of the clean area/contaminated area, and vehicle movements which could potentially be the cause of ASFV entry into the farm.

It is also necessary to significantly expand this part of the article by adding data on the spread of the virus in the environment (specific features in the region being described).

The spread of the virus in the environment was detailed in Figure 1 and 2. Figure 2 was modified according to the revisions suggested to the other Reviewer. We trust that the changes will help clarify the Italian ASF epidemiology.

The safety of the virus in the environment when pigs are free-grazing, and in water if it does not undergo sanitary cleaning, also requires clarification.

In Italy, free-grazing pigs are not allowed as this practice is illegal. However, there are backyard farms (private swine farms) in some areas of Italy. As reported in the article “all private swine farms in Pavia province were closed”. Anyway, these farms are required to comply with Italian and European legislation on biosecurity in pig farms. The legislation has been added to the text in the introduction section.

There are no described cases of burial of pig bodies as a source of re-infection (and it is not clear whether such cases are not observed or are simply not described enough).

In Italy, the safety disposal of carcasses was carried out via rendering (no burial). The Authors have been now added this information in par. 2.3.1. 

In other words, the article needs to pay more attention to possible routes of transmission of the virus in the region and their characteristics (if any).

The authors sincerely thank the reviewer for their input, which has enriched the article with additional information and insights on the strategic challenges to eradicate ASF in Italy.